# Management and outcomes of gastrointestinal congenital anomalies in low, middle and high income countries: protocol for a multicentre, international, prospective cohort study

Naomi Jane Wright, Global PaedSurg Research Collaboration

King's Centre for Global Health and Health Partnerships, King's College London, London, UK

**Correspondence to**
Naomi Jane Wright;
naomiwright@doctors.org.uk

## ABSTRACT

**Introduction** Congenital anomalies are the fifth leading cause of death in children <5 years of age globally, contributing an estimated half a million deaths per year. Very limited literature exists from low and middle income countries (LMICs) where most of these deaths occur. The Global PaedSurg Research Collaboration aims to undertake the first multicentre, international, prospective cohort study of a selection of common congenital anomalies comparing management and outcomes between low, middle and high income countries (HICs) globally.

**Methods and analysis** The Global PaedSurg Research Collaboration consists of surgeons, paediatricians, anaesthetists and allied healthcare professionals involved in the surgical care of children globally. Collaborators will prospectively collect observational data on consecutive patients presenting for the first time, with one of seven common congenital anomalies (oesophageal atresia, congenital diaphragmatic hernia, intestinal atresia, gastroschisis, exomphalos, anorectal malformation and Hirschsprung's disease).

Patient recruitment will be for a minimum of 1 month from October 2018 to April 2019 with a 30-day post-primary intervention follow-up period. Anonymous data will be collected on patient demographics, clinical status, interventions and outcomes using REDCap. Collaborators will complete a survey regarding the resources and facilities for neonatal and paediatric surgery at their centre.

The primary outcome is all-cause in-hospital mortality. Secondary outcomes include the occurrence of postoperative complications. Chi-squared analysis will be used to compare mortality between LMICs and HICs. Multilevel, multivariate logistic regression analysis will be undertaken to identify patient-level and hospital-level factors affecting outcomes with adjustment for confounding factors.

**Ethics and dissemination** At the host centre, this study is classified as an audit not requiring ethical approval. All participating collaborators have gained local approval in accordance with their institutional ethical regulations. Collaborators will be encouraged to present the results locally, nationally and internationally. The results will be submitted for open access publication in a peer reviewed journal.

**Trial registration number** NCT03666767

## Strengths and limitations of this study

► This will be the first large-series, geographically comprehensive, multicentre, international, prospective cohort study to define the management and outcomes of a selection of common congenital anomalies in low, middle and high income countries across the globe.
► The collaborative approach for this study allows a large series of high-quality data to be collected in a timely manner without overburdening high-volume, low-resource centres.
► The seven study conditions constitute a selection of the most common life-threatening congenital anomalies requiring emergency surgical care in the neonatal period (box 1).
► We recognise that some children may not reach a facility capable of providing acute paediatric surgical care and hence the results obtained may be an underestimation of true morbidity and mortality, especially in low and middle income countries.
► The number of variables being collected per patient has been limited to those known to have the greatest impact on outcomes to optimise the feasibility of the study; follow-up is limited to 30 days post-primary intervention.

## INTRODUCTION

In 2015, the Global Burden of Disease study concluded congenital anomalies (also known as congenital malformations, congenital abnormalities or birth defects) to be the fifth leading cause of death in children <5 years of age globally.[1] This equates to approximately half a million deaths from congenital anomalies each year, 97% of which occur in low and middle income countries (LMICs). Indeed, this is likely to be an underestimation of the actual number of deaths due to underdiagnosis of neonates with congenital anomalies

---

**Box 1  Congenital anomalies in the Global PaedSurg Study**

- ► Oesophageal atresia ± tracheo-oesophageal fistula
- ► Congenital diaphragmatic hernia
- ► Intestinal atresia
- ► Gastroschisis
- ► Exomphalos
- ► Anorectal malformation
- ► Hirschsprung's disease

who die in the community and a lack of death certification in many LMICs.[2] Not only is the mortality rate higher in LMICs, but the prevalence is also higher due to micronutrient deficiencies, infections and teratogens during pregnancy resulting in more cases and a lack of antenatal diagnosis prohibiting terminations.[3 4] There is limited research and a lack of congenital anomaly registries in LMICs, and hence they have received very little global attention.[5]

The conditions forming the focus of this study (box 1) constitute a selection of the most common life-threatening congenital anomalies during the neonatal period, which involve the gastrointestinal tract. They each have an incidence of 1/2000 to 1/5000, they collectively form up to 40% of emergency neonatal surgery and associated mortality can be in excess of 50% in many LMICs.[6–9] Disparities in outcomes globally can be stark; for example, the mortality from gastroschisis is 75%–100% in many LMICs compared with 4% or less in high income countries (HICs).[10–12] Reasons for poor outcomes include a lack of antenatal diagnosis, delayed presentation, limited neonatal transport and in-hospital resources, a dearth of trained support personnel and a lack of intensive care and parenteral nutrition for neonates.[9 13 14] In Uganda, it was calculated that only 3.5% of the need for neonatal surgery was met by the healthcare system.[8]

In 2010, the World Health Assembly passed a resolution recommending 'prevention whenever possible, to implement screening programmes and to provide care and ongoing support to children with birth defects and their families'.[2] Prevention is paramount; however, this is not yet possible for many congenital anomalies and hence a focus on improving postnatal care and outcomes is vital. The Sustainable Development Goal 3.2 aims to end preventable deaths of newborns and children under the age of 5 years by 2030.[6 15 16] With one-third of infant deaths being attributed to congenital anomalies, clearly, this will not be achievable without an accelerated effort towards the provision of surgical care for children. It is estimated that two-thirds of deaths and disability from congenital anomalies can be avoided with the provision of neonatal and paediatric surgical care.[6] Indeed, studies have demonstrated such provision can be highly cost-effective in terms of disability-adjusted life-years saved.[5] Yet, neonatal and paediatric surgical care remains a low priority on the global health agenda.[5]

A shift is needed to focus on the provision of surgical care for children within National Health Plans and International Organisations and to elevate congenital anomalies on the global health agenda. This large-scale, geographically comprehensive, multicentre prospective cohort study aims to define the current management and outcomes of a selection of common congenital anomalies globally and identify factors affecting outcomes that can be modified to improve care. This is vital to aid advocacy and global health prioritisation and inform future interventional studies aimed at improving outcomes.

## AIM

To undertake the first large-scale, geographically comprehensive multicentre, prospective cohort study comparing the management and outcomes of a selection of common congenital anomalies in low, middle and high income countries across the globe.

## OBJECTIVES

1. To compare the mortality and post-intervention complications of a selection of common congenital anomalies involving the gastrointestinal tract in LMICs and HICs globally.
2. To identify patient-level and hospital-level factors affecting outcomes that be modified to improve care.
3. To establish a research collaboration consisting of children's surgical care providers across the world to help enhance research capacity and to create a platform for ongoing collaborative research and intervention studies aimed at improving outcomes.
4. To raise awareness and provide advocacy for neonatal and paediatric surgical care within global health prioritisation, planning, policy and funding.

## METHODS AND ANALYSIS
### Study design

This is an international, multicentre, prospective observational cohort study. The Global PaedSurg Research Collaboration consisting of children's surgical care providers (collaborators) across the world was established from November 2017 to co-ordinate the study at an institutional level and facilitate data collection. Collaborators are free to choose one or more months between 1 October 2018 to 30 April 2019 (inclusive) to recruit consecutive patients to the study, with a 30-day post-primary intervention follow-up period. The primary intervention must occur within 30 days of presentation to be included in the study. Hence, the last date for primary data collection is 29 June 2019. Following this, there will be a period of data collection for the data validation process continuing until the end of August 2019.

### Collaborators

International collaborators will have a variety of roles and responsibilities within the study. Local collaborators

will establish mini-teams locally, gain study approval, use the protocol criteria to appropriately identify patients for study inclusion, collect prospective data and upload it to the Research Electronic Data Capture (REDCap) online system. Each hospital will have a local study lead who will hold overall responsibility for ensuring the data are accurate, complete and without duplications. Country-lead collaborators will help to recruit other collaborators from within their country and provide advice and support regarding gaining local study approval and data collection. They may also help with translation of the study literature to the local language, if required. Continent and regional leads will help to recruit country leads, provide them with advice regarding the study and also encourage and co-ordinate presentations of the protocol at national and international meetings. Lead investigators contributed to the study design through the provision of feedback from the pilot studies undertaken in multiple languages. An organising committee will help to co-ordinate all study activities and a steering committee will provide guidance throughout.

There are a number of benefits for collaborators participating in the study. Publishing journal(s) will be asked to make all collaborators PubMed-citable co-authors. This is based on an equal partnership model described by the Lancet and is used by a number of national and international collaboratives.[17–21] All collaborators will be listed as an author on resulting presentations. Collaborators will have the opportunity to present the study locally, nationally and internationally, initially the study protocol and later the results. This often provides collaborators, especially those who are junior or from LMICs, the opportunity to apply for funding to attend, present and network at such meetings. Participation in the study provides an easy route and insight into clinical research, which can be further established through participation in the 2-year Research Training Fellowship that is running alongside the main study free of charge for all interested collaborators.

## Sample selection
### Collaborator and hospital inclusion criteria
All hospitals and healthcare professionals providing surgical care for neonates and children, presenting for the first time, with one or more of the study conditions can be included in the study. Collaborators should gain permission from the senior surgeon or physician who oversees the care of the children to be included in the study in order to participate. There can be up to three collaborators in a mini-team per month of data collection. One mini-team can collect data over one or more months or several mini-teams can collect data over a different month each. Each mini-team must contain at least one senior surgeon or physician to oversee the data collection process.

### Patient inclusion and exclusion criteria
Any neonate, infant or child under the age of 16 years, presenting acutely for the first time, with one or more of the study conditions can be included in the study.

Patients who have previously received surgery for their presenting condition or those representing with a complication of surgery are excluded. Patients presenting electively for surgery are excluded. Children who have received basic resuscitative care for their condition at a different healthcare facility and are then transferred to the study centre for their primary surgical intervention can be included. Children who only receive resuscitative treatment at the study centre and are then referred elsewhere for their primary surgical intervention cannot be included since the outcome of the surgical care will not be known and also to avoid the risk of duplicate patients in the study. Patients who receive conservative treatment as their primary intervention, palliative care or no care must be included within the study to accurately reflect the management and outcomes of all presenting cases.

If a patient presents with more than one of the study conditions, the details of each condition that they present acutely with can be included, but not a previously managed condition. For example, a newborn presenting with oesophageal atresia and anorectal malformation would have both conditions included. A patient presenting for the first time with Hirschsprung's disease at several months of age who had a duodenal atresia repaired at birth would have the full details of the Hirschsprungs disease included, but the duodenal atresia would simply be noted as an associated anomaly.

## Outcome measures
The primary outcome is all-cause, in-hospital mortality.

For patient's hospitalised for >30 days following primary intervention, a 30-day post-primary intervention mortality rate will be used. Those who do not receive a primary intervention, but remain alive and hospitalised at 30 days following primary admission, will have this time point used for recording their mortality status for the primary outcome. Primary outcome is defined in table 1.

The secondary outcomes include complications occurring within 30 days of primary intervention:
► Surgical-site infection.
► Wound dehiscence.
► Need for re-intervention.
► Condition-specific complications.
► Condition specific outcome variables.
► Length of hospital stay or time from admission to death in patients who do not survive.
► 30-day post-primary intervention mortality.

Secondary outcomes will not be collected on patients who do not receive a primary intervention within 30 days of hospital admission, with the exception of length of hospital stay or time from admission to death. Thirty-day follow-up will be undertaken within the capacity of the collaborating team; no additional funding will be provided.

## Data collection
Generic variables relating to the patient demographics, antenatal care, prehospital care, clinical condition, surgical

**Table 1** Generic data points

| Generic questions | Answers |
|---|---|
| During which month did the patient present to your hospital? | October, November, December, January, February, March, April.<br>Please select the month that the patient presented to your hospital for the first time with this congenital anomaly. For example, if a baby was born with gastroschisis on the 29 September and presented to your hospital on 1 October you should select October. |
| Has consent been provided to include this patient in the study?<br>If no, which condition did the patient present with? | Yes, No, Patient consent is not required for this study at my institution.<br><br>Oesophageal atresia, Congenital diaphragmatic hernia, Intestinal atresia, Gastroschisis, Exomphalos, Omphalocele, Anorectal malformation, Hirschsprung's Disease.<br>Please select all the conditions that the patient presented with. Do not select a condition which the patient has already received surgical treatment for previously. |
| **Demographics** | |
| Gestational age at birth | Number of weeks from the first day of the women's last menstrual cycle until birth. Round up or down to the nearest week. |
| Age at presentation (in hours) | We understand this information may be difficult to obtain - please be as accurate as you can. Please round to the nearest hour. This number may be very large for patients who have a delayed presentation - please still enter it. For neonates born within your centre please enter 0. Enter unknown if unknown. |
| Gender | Male, Female, Ambiguous, Unknown. |
| Weight at presentation | In kilograms (kg) on the day of presentation. Please provide a value to one decimal place. |
| Does the patient have another anomaly in addition to the study condition? | Yes: Cardiovascular, Yes: Respiratory, Yes: Gastrointestinal, Yes: Neurological, Yes: Genito-urinary, Yes: Musculoskeletal, Yes: Down syndrome, Yes: Beckwith-Wiedemann syndrome, Yes: Cystic fibrosis, Yes: Chromosomal, Yes: Other, No.<br>Select all that apply. Include all anomalies diagnosed at any stage up until 30 days post-primary intervention or 30 days following presentation for those who did not receive an intervention. If you suspect an associated anomaly, but it has yet to be diagnosed, select 'Yes: Other'. |
| Distance from the patient's home to your hospital | In kilometres (km). Please round to the nearest kilometre. Please enter 0 if born in your hospital. |
| **Antenatal care and delivery** | |
| Antenatal ultrasound undertaken?<br>If the condition was diagnosed antenatally, at what gestational age? | Yes: study condition diagnosed, Yes: problem identified but study condition not diagnosed, Yes: no problem identified, No.<br><br>Please round up to the nearest week. If the patient has more than one study condition, please note the gestational age at which one or more of the conditions was first diagnosed. |
| Mode of transport to hospital?<br>Where did the patient present from? If other, please specify. | Ambulance, Other transport provided by the health service, Patient's own transport, Born within the hospital.<br><br>Home, Community Clinic, General Practice, District Hospital, Other, Unknown.<br>District hospital includes secondary-level healthcare, provincial hospital, general hospital, general mission hospital or regional hospital. It has general anaesthesia and can provide general surgical care. |
| Type of delivery: | Vaginal (spontaneous), Vaginal (induced), Caesarean section (elective), Caesarean section (urgent/non-elective), Unknown. Vaginal delivery includes those requiring forceps and ventouse. |
| **Clinical condition and patient care** | |
| Was the patient septic on arrival?<br><br><br>If yes, were appropriate antibiotics administered? | Yes, no.<br>Sepsis is SIRS with a suspected or confirmed bacterial, viral or fungal cause. SIRS is a response to a stimulus, which results in two or more of the following: temperature >38.5°C or <36°C, tachycardia*, bradycardia* in children <1 year old, tachypnoea*, leucopenia or leucocytosis*, hyperglycaemia*, altered mental status, hyperlactaemia*, increased central capillary refill time >2 seconds. Arrival is the time of birth for neonates born at your hospital. *Variables are defined as values outside the normal range for age.<br>Yes: within 1 hour of arrival, Yes: within the first day of arrival, No.<br>Appropriate antibiotics are defined as either broad spectrum covering gram negative, gram positive and anaerobic bacteria OR antibiotics that are the standard empirical treatment for that condition according to local guidelines OR are based on sensitivities provided by a microbiology sample. |

Continued

**Table 1** Continued

| Generic questions | Answers |
|---|---|
| Was the patient hypovolaemic on arrival?<br>If yes, was an intravenous fluid bolus given?<br>If yes, how much intravenous fluid was given? | Yes, No. Criteria for diagnosis include at least one of the following: prolonged central capillary refill time >2 seconds, *tachycardia, mottled skin, *reduced urine output, cyanosis, impaired consciousness, *hypotension. *Variables are defined as values outside the normal range for age.<br><br>Yes: within 1 hour of arrival, Yes: on the first day of arrival, No.<br><br>10–20 mL/kg, >20 mL/kg.<br>If <10 mL/kg was given, please select 'no' for the question asking if intravenous fluid was given. |
| Was the patient hypothermic on arrival?<br>If yes, was the patient warmed on arrival to within a normal temperature range? | Yes, No. Defined as <36.5°C core temperature. Arrival is the time of birth for neonates born at your hospital.<br><br>Yes, No. Only select yes if warming was commenced within 1 hour of arrival. Arrival is the time of birth for neonates born at your hospital. |
| Did the patient receive central venous access?<br>If yes, did the patient acquire central line sepsis? | Yes: umbilical catheter, Yes: peripherally inserted central catheter, Yes: percutaneously inserted central line with ultrasound guidance, Yes: surgically placed central line (open insertion), No.<br>Please select all that the patient received within 30 days of primary intervention or 30 days of presentation if no intervention was undertaken.<br><br>Yes: diagnosed clinically, Yes: confirmed on microbiology, No.<br>Within 30 days of primary intervention or 30 days of presentation if no intervention was undertaken. |
| Time from arrival at your hospital to primary intervention in hours | Enter 0 if no intervention was undertaken.<br>*Primary intervention for each condition is defined as follows. Oesophageal atresia: surgery, either temporising or definitive, to manage the oesophageal atresia and/or tracheo-oesophageal fistula. Congenital diaphragmatic hernia: surgery to reduce the hernia and close the defect. Intestinal atresia: surgery, either temporising or definitive, to manage the obstruction including stoma formation and primary anastomosis. Gastroschisis: any procedure to either cover or reduce the bowel and/or close the defect. This includes application of a silo (regardless of whether or not they go on to require surgery). It excludes initial covering of the bowel in a plastic covering (bag or cling film) prior to intervention. Exomphalos: surgery or application of topical treatment to the sac in patients managed conservatively (regardless of whether or not they go on to require surgery). Hirschsprung's disease: surgery, either temporising or definitive, or rectal/distal bowel irrigation, laxatives or digital stimulation in patients managed conservatively. This does not include pre-operative washouts in patients planned to have surgery. Anorectal malformation: surgery, either temporising or definitive, or anal/fistula dilatation in patients with a low anorectal malformation managed conservatively.* |
| American Society of Anesthesiologists Score at the time of primary intervention | 1.Healthy person, 2. Mild systemic disease, 3. Severe systemic disease, 4. Severe systemic disease that is a constant threat to life, 5. A moribund patient who is not expected to survive without the operation, Not applicable — no intervention. |
| What type of anaesthesia was used for the primary intervention? | General anaesthesia with endotracheal tube, General anaesthesia with laryngeal airway, Ketamine anaesthesia, Spinal/caudal anaesthesia, Local anaesthesia only, No anaesthesia/just analgesia, No anaesthesia/no analgesia, Not applicable: no surgery or intervention undertaken. |
| Who undertook the anaesthetic for the primary intervention? | Anaesthetic doctor, Anaesthetic nurse, Medical officer, Surgeon, Other healthcare professional, No anaesthetic undertaken.<br>If more than one of these personnel were present, please select the most senior. |
| Who undertook the primary intervention? | Paediatric surgeon (or junior with paediatric surgeon assisting/in the room), General surgeon (or junior with paediatric surgeon assisting/in the room), Junior doctor, medical officer or other (without a paediatric or general surgeon assisting/in the room), Trainee surgeon (without a paediatric or general surgeon assisting or in the room), Not applicable — no surgery or primary intervention undertaken. |
| Was a Surgical Safety Checklist used at the time of primary intervention? | Yes, No: but it was available, No: it was not available, Not applicable: a conservative primary intervention was undertaken, Not applicable: no surgery or primary intervention undertaken. |
| Total duration of antibiotics following primary intervention | In days (including the day of surgery and the day antibiotics were stopped. Include intravenous and oral antibiotics). |
| Did the patient receive a blood transfusion? | Yes: not cross-matched, Yes: cross-matched, No: not required, No: it was required but not available.<br>Within 30 days of primary intervention or 30 days of presentation if no intervention was undertaken. |
| Did the patient require ventilation?<br>If yes, for how long did the patient remain on ventilation? | Yes: and it was given, Yes: but it was not available, No.<br>Within 30 days of primary intervention or 30 days of presentation if no intervention was undertaken. Please include all types of ventilation.<br><br>In days (include all days on ventilation within 30 days of primary intervention or 30 days of presentation if no intervention was undertaken). |

### Table 1 Continued

| Generic questions | Answers |
|---|---|
| Time to first enteral feed (post-primary intervention) | In days (include the day of primary intervention and the day of first enteral feed in the calculation). Enter 0 if enteral feeds were not commenced. Enter 999 if feeds were not stopped, for example, in patients with Hirschsprung's Disease managed conservatively. Include all types of enteral feeding — oral, nasogastric, gastrostomy and other. |
| Time to full enteral feeds (post-primary intervention) | In days (enter 0 if the patient died before reaching full enteral feeds or 30 if the patient had not reached full enteral feeds at 30 days post-primary intervention or 30 days following admission in patients who did not receive a primary intervention). Include all types of enteral feeding — oral, nasogastric, gastrostomy and other. |
| Did the patient require parenteral nutrition? If yes, for how long did the patient receive parenteral nutrition? | Yes and it was given, Yes and it was sometimes available but less than required, Yes but it was not available, No. <br><br>In days. Include all days that the patient received parenteral nutrition (any volume) up until 30 days post-primary intervention or 30 days following presentation in patients who do not receive an intervention. |
| **Outcomes** | |
| Did the patient survive to discharge? If the patient was discharged prior, were they still alive at 30 days following primary intervention? If no, cause of death? | Yes, No. <br>Select yes if the patient was still alive in your hospital 30 days after primary intervention or 30 days after presentation in patients who did not receive a primary intervention. <br><br>Yes, No: not followed-up after discharge, Followed-up but not until 30 days post-primary intervention. This can include all reliable communication with the patient/patient's family including in person, via telephone and other. <br><br>Sepsis, aspiration pneumonia, respiratory failure, cardiac failure, malnutrition, electrolyte disturbance, haemorrhage, lack of intravenous access, hypoglycaemia, recurrent tracheo-oesophageal fistula, recurrent diaphragmatic hernia, anastomotic leak, ischaemic bowel, ruptured exomphalos sac, enterocolitis, other. If other, please specify. |
| Duration of hospital stay (days) | Please include the day of admission and the day of discharge in your calculation. For example, if a patient presented on 1 October and was discharged on the 5 October, their duration of hospital stay would be 5 days. If the patient died, please record the number of days from admission to death. Only include the duration of the primary admission, not subsequent admissions if the patient re-presented. |
| Did the patient have a surgical site infection? | Yes, No, Not applicable: no surgical wound. <br>This is defined as one or more of the following within 30 days of surgery: (1) purulent drainage from the superficial or deep (fascia or muscle) incision, but not within the organ/space component of the surgical site OR (2) at least two of: pain or tenderness, localised swelling, redness, heat, fever, AND the incision is opened deliberately to manage infection, spontaneously dehisces or the clinician diagnoses an SSI (negative culture swab excludes this criterion) OR (3) there is an abscess within the wound (clinically or radiologically detected). |
| Did the patient have a full thickness wound dehiscence? | Yes, No, Not applicable — no surgical wound. <br>This is defined as all layers of the wound opening within 30 days of surgery. |
| Did the patient require a further unplanned intervention? | Yes — percutaneous intervention, Yes — surgical intervention, No, Not applicable — no primary intervention undertaken. <br>Within 30 days of primary intervention. This does not include routine reduction and closure of the defect in neonates with gastroschisis receiving a preformed silo. |
| Was the patient followed up at 30 days post primary surgery or intervention to assess for complications? | Yes: reviewed in person, Yes: via telephone consultation, Yes: via other means, Yes: still an in-patient at 30 days, No: data are based on in-patient observations only, No: follow-up was done but prior to 30 days. |
| If the patient had a complication, when was it diagnosed? | During the primary admission, As an emergency re-attender, At routine follow-up as an outpatient, Not applicable: no complications. |
| What study condition does this patient have? | Oesophageal atresia, Congenital diaphragmatic hernia, Intestinal atresia, Gastroschisis, Exomphalos/Omphalocele, Anorectal malformation, Hirschsprung's Disease. <br>If the patient has presented for the first time with more than one of these conditions, please select all that apply. If the patient presented on this occasion with one of these conditions, but previously had another condition managed then only select the condition they are presenting with on this occasion and enter that they have another anomaly in the demographics section above. For example, if the patient presents at 2 months with Hirschsprung's disease, but previously had a duodenal atresia repair, please select Hirschsprung's disease here (not intestinal atresia) and tick in the section above that they have another gastrointestinal anomaly. |

SIRS, Systemic Inflammatory Response Syndrome.

intervention and outcomes will be collected for all patients in the study (table 1). Specific variables will be collected for each individual condition (online supplementary file 1).

Outcomes and variables have been chosen using published core outcome sets and commonly collected outcomes in systematic reviews and meta-analyses.[22–37] Collaborators will enter anonymous, deidentified data via the secure internet-based REDCap system. This will be stored on King's College London REDCap server.

A short survey will be completed by the local study lead and one other collaborating consultant or registrar on the resources and facilities available for neonatal and paediatric surgical care at their centre (online supplementary file 2).

## Data quality

To ensure high quality of data, a detailed protocol for collaborators has been produced and published on the study website (www.globalpaedsurg.com) in 12 languages: English, French, Spanish, Portuguese, German, Italian, Chinese, Arabic, Korean, Lithuanian, Turkish and Russian. Clear and concise definitions have been provided for all data points on the protocol, on the data collection forms and within REDCap when entering the data. A study launch meeting was undertaken where the principal investigator presented the data collection process in detail, demonstrated use of REDCap and answered questions. This was recorded, circulated to all collaborators via email and placed on the website. A frequently asked questions document has been circulated via email and placed on the website. Two meetings were held by the principal investigator to detail the study, data collection process and answer questions among the country leads so they in turn can provide advice and support to local collaborators within their country. Again this was recorded, circulated and placed on the website.

A pilot study of the patient data collection form and institutional survey was undertaken by lead investigators to optimise the study design and to address any feasibility or other barriers to effective data collection and study completion across participating sites. The pilot study commenced on 1 August 2018 for 30 days in English, Spanish and French by 41 collaborator colleagues. The data collection forms were amended following feedback to clarify terminology, add important missing variables or descriptions and correct any translation errors. All translated data collection forms, REDCap and study documentation have been checked and verified by a native speaker for accuracy.

## Data validation

Ten percent of collaborating centres will be selected at random for data validation by an independent research collaborator. The aim will be to determine the numbers of patients eligible during the data collection period to check if any were missed and collect a selection of data again to cross-check for accuracy. Validating questions have been built into the data collection tool. At least 90% of primary and secondary outcomes must be completed

for each patient. All collaborators within validating centres will be asked to complete a brief survey regarding their experience with data collection to identify any potential errors and to aid with data interpretation.

## Sample size calculation

A sample size calculation was undertaken using Stata/IC V.15.0 based on Bonferroni correction for multiple testing, assuming 80% power and an overall type 1 error of 5%. The required sample size for each condition has been calculated for the primary outcome of mortality in LMICs compared with HICs and also low, middle and high income countries separately (table 2). Mortality estimations are based on pooled data from published studies on these conditions in low, middle and high income countries, respectively.

Based on the patient numbers included in the previously undertaken PaedSurg Africa study, which used a similar study design, the estimated sample sizes to detect a significant difference between LMICs and HICs in this study are achievable.[11]

## Estimated study population

The mean number of cases presenting to an institution per month for each study condition was estimated from published studies across all income settings (table 2). On average, most institutions caring for patients with these conditions receive 1–2 new cases per month; each participating institution would expect approximately 7–14 new cases in the study per month although this can vary. The aim is to include a minimum of 365 months of data; 183 months from LMICs and 183 months from HICs. This should ensure enough cases of exomphalos to determine a significant difference between LMICs and HICs; fewer months of data are required to determine significant differences between other study conditions. An up-to-date total of patient numbers within the study will be maintained on the study website.

## Data analysis
### Patient and institutional data

Data will be analysed using Stata and SAS V.9.4 (Cary, North Carolina, USA). Missing data for the covariates will be analysed to determine whether it is related to the outcome and either complete-case analyses or multiple imputation techniques will be used for analyses accordingly.

Significant differences in mortality between LMICs and HICs will be determined for each of the study conditions using $X^2$ analysis, or Fischer's exact test if either group contains <10 patients. World Bank classification of low, middle and high income countries during the fiscal year 2018 will be used.[38]

Univariate logistic regression analyses will be conducted between covariates and the primary outcome of mortality. Based on the results, covariates with a p value <0.10 will be included in the multivariate model. The final multilevel multivariate logistic model will be determined using

**Table 2** Estimated mortality and sample sizes for low, middle and high income countries and the mean number of cases per month per institution globally

| Condition | Mortality LIC (%, n) | Mortality MIC (%, n) | Mortality LMIC combined (%, n) | Mortality HIC (%, n) | Sample size for LIC | Sample size for MIC | Sample size for HIC | Sample size for LMIC vs HIC (per group) | Mean no. cases/ month/ institution (L, M and HIC combined) |
|---|---|---|---|---|---|---|---|---|---|
| OA ±TOF | 79.5% (62/78) | 41.8% (623/1488) | 43.7% (685/1566) | 2.7% (6/221) | 34 | 34 | 23 | 21 | 1.02 |
| CDH | – | 47.4% (130/274) | 47.4% (130/274) | 20.4% (201/982) | – | – | – | 63 | 0.54 |
| IA | 42.9% (42/98) | 40.0% (97/241) | 41.0% (139/339) | 2.9% (12/407) | 6014 | 6014 | 25 | 24 | 0.63 |
| Gastroschisis | 83.1% (211/254) | 42.6% (205/481) | 56.6% (416/735) | 3.7% (28/748) | 29 | 29 | 24 | 15 | 0.85 |
| Exomphalos | 25.5% (41/161) | 31.9% (132/414) | 30.1% (173/575) | 12.7% (40/316) | 1040 | 1040 | 196 | 115 | 0.63 |
| ARM | 26.3% (26/99) | 17.5% (243/1391) | 18.1% (269/1490) | 3% (14/462) | 460 | 460 | 90 | 85 | 1.34 |
| Hirschsprung's disease | 19.1% (33/173) | 16.8% (55/328) | 17.6% (88/501) | 2.3% (43/1897) | 5802 | 5802 | 85 | 79 | 2.21 |

ARM, anorectal malformation; CDH, congenital diaphragmatic hernia; HIC, high income countries; IA, intestinal atresia; LIC, low income countries; LMIC, low and middle income countries; MIC, middle income countries; OA, oesophageal atresia; TOF, tracheo-oesophageal fistula.

stepwise backward elimination to interventions and peri-operative factors affecting outcomes. Data will be adjusted for confounding factors and effect modifiers. Potential confounders include gestation age at birth, weight, time from birth to presentation and American Society of Anesthesiologists score at the time of primary intervention. Potential effect modifiers include administration of peri-operative antibiotics, fluid resuscitation, thermal control and provision of other condition-specific neonatal care such as parenteral nutrition in neonates with gastroschisis.

Multi-level multivariate logistic regression analysis will also be undertaken to identify institutional factors affecting mortality with adjustment for confounders. $P<0.05$ will be deemed significant.

### Data validation

A weighted kappa statistic will be used to determine the level of agreement between the patient data in the main study and the validation data. It will also be used to determine the level of agreement between institutional surveys independently completed by the local study lead and one other consultant or registrar at each participating centre. Results will be presented as a proportion of agreement for each variable being validated.

### Patient and public involvement

CDH UK, a patient and family advisory group and charity, provided input into the design of the study protocol and data collection tool. Their input will be sought on the findings and dissemination of the results.

## ETHICS AND DISSEMINATION
### Research ethics approval

The study has been classified as an audit at the host institution and hence did not require ethical approval. The study fulfils the audit criteria as follows: (1) All data collected measures current practice. The study does not involve any changes to patient management. (2) Current practice and outcomes in low, middle and high income countries will be compared with published standards in the literature. Table 2 details the current mortality standards for each of the seven study conditions in high income countries. (3) All the study data are routinely collected information which should be known to the study team without asking additional questions to the patients/parents. (4) All data to be entered into REDCap are entirely anonymous. (5) No individual patient, collaborator, institution or country will be independently identifiable in the study results. (6) All data will be stored securely and will be governed by King's College London data protection team.

Research collaborators were required to gain approval to participate in the study at their institution according to their local ethical regulations. Data transfer agreements were legally signed between institutions where required. The participating institutions, type of study approval and study approval reference numbers are detailed in online supplementary file 3. It was not mandated for study approvals to be translated into English. Hence, some reference numbers are in the local scripture of the participating country and have therefore not been incorporated into the table.

## Study dissemination

The study concept and design will be presented at international conferences in order to recruit collaborators. Following completion, the results will be presented at local, national and international conferences globally. Both the promotional presentations of the study protocol and the study results will be presented by study collaborators of all levels of training, disciplines and regions of the world. The results will be submitted for open access publication in a peer reviewed journal. Following publication, the full anonymous, deidentified dataset will be made publicly available via an online repository. Collaborators will have the opportunity to undertake sub-analyses of the data for their country (if all collaborators from that country agree), region or continent.

## DISCUSSION

This study aims to define, for the first time, the management and outcomes of a selection of common life-threatening congenital anomalies across the globe. This will help to raise awareness of the unacceptable disparities in outcomes between low, middle and high income countries and the need to focus on improving access to quality surgical care for neonates with congenital anomalies within national health plans and global health prioritisation. It is hoped that factors affecting mortality and morbidity will be identified that can be modified to improve care. Establishment of the Global PaedSurg Research Collaboration developed during this study will create a platform for ongoing collaborative work and interventional studies aimed at improving outcomes in the future.

**Twitter** @GlobalPaedSurg

**Acknowledgements** Thank you to Bolaji Coker for the REDCap administration and management. Thank you to Beverley Power (CDH UK) for representing patients, parents and families through your feedback on the study design during the pilot study. Thank you to Xiya Ma and Dylan Goh for helping with the Chinese translation of study documentation.

**Collaborators** Principal Investigator: Naomi Wright (King's College London, UK). Steering Committee: Niyi Ade-Ajayi (King's College Hospital, UK), Adesoji Ademuyiwa (Lagos University Teaching Hospital, Nigeria), Emmanuel Ameh (National Hospital, Abuja, Nigeria), Justine Davies (University of Birmingham, UK), Kokila Lakhoo (University of Oxford and Oxford University Hospitals, UK), Dan Poenaru (McGill University, Montreal, Canada), Nick Sevdalis (King's College London, UK), Emily Smith (Baylor University, Texas, USA), Andrew Leather (King's College London, UK). Writing Committee: Harmony Ubhi (King's College London, UK), Samuel Parker (Imperial College London, UK), Godfrey Sama Philipo (Muhimbili University of Health and Allied Sciences, Tanzania). Organising Committee: Sadi Abukhalaf (Al Quds University, Palestine), Nana Adofo-Ansong (Mahikeng Provincial Hospital, South Africa), Melika Akhbari (King's College London, UK), Ahmad Alhamid (University of Aleppo, Syria), Osaid H. Alser (University of Oxford, UK), Emrah Aydin (Koc University, Turkey), Yousra-Imane Benaskeur (Universite de Montreal, Canada), Shrouk M. Elghazaly (Assiut University, Egypt), Safa abdal Elrais (University of Tripoli, Libya), Sophia Hashim (University College London, UK), Laura Herrera (Geisel School of Medicine, Dartmouth, USA), Gabriella Hyman (University of Witwatersrand, South Africa), Henang Kwasau (College of Medicine and Allied Health Sciences, University of Sierra Leone), Yang Liu (Children's Hospital, Zhejiang University School of Medicine, China), Bruno Martinez-Leo (Moctezuma Children's Hospital, Mexico), Kelly Naranjo (Columbia University Medical Centre, USA), Ibrahim Nour (Jordan University Hospital, Jordan), Cristiana Riboni (University of Pavia, Italy), Mahmoud Saleh (University of Gezira, Sudan), Hosni Khairy Salem (Cairo University, Egypt), Patricia Shinondo (University Teaching Hospital, Lusaka, Zambia), Marcus Sim (Stepping Hill Hospital, UK), Hannah Thompson (King's College Hospital, UK), Agota Vaitkiene (Vilnius University Hospital Santaros Kliniko, Lithuania), Dominique Vervoort (Harvard Medical School, USA), Isabelle Williams (Cambridge University, UK), Aayenah Yunus (King's College London, UK). Lead Investigators: Muhammad Amjad Chaudhary, Muhammad Adnan Khan Khattak, Muhammad Bin Amjad (Children's Hospital, PIMS, Islamabad, Pakistan), Marlene Dominguez Anaya (Children's Hospital Manuel A.Villarroel Cochabamba, Bolivia), Samiul Hasan, Sabbir Karim, Ashrarur Rahman Mitul (Dhaka Shishu (Children) Hospital, Bangladesh), Paolo Bragagnini, Segundo Rite (Hospital Universitario Miguel Servent, Zaragoza, Spain), Hana Arbab, Lubna Samad, Aqil Soomro (Indus Hospital, Pakistan), Niveshni Maistry (John Radcliffe Hospital, UK), Raed Nael Al-Taher, Ibrahim Rabi Nour, Osama Abdul Kareem Sarhan (Jordan University Hospital, Jordan), Muhammad Arshad, Taimur Qureshi, Hina Yousaf (Liaquat National Hospital, Pakistan), Candy SC Choo, Doris Mae Dimatatac, Shireen Anne Nah (KK Women's and Children's Hospital, Singapore), Vijay Anand Ismavel, Ann Miriam, Shajin T (Makunda Christian Leprosy and General Hospital, India), Monica Ivanov, Andreea Serban (Marie Curie Hospital in Bucharest, Romania), Bruno Martinez-Leo (Moctezuma Children's Hospital, Mexico), Eva Blazquez-Gomez, Luis Garcia-Aparicio, Martí Iriondo, Jordi Prat, Xavier Tarrado (Hospital Sant Joan de Deu, Spain), Lars Hagander, Emma Svensson (Skane University Hospital's Pediatric Care Hospital, Lund, Sweden), Alhassan Abdul-Mumin, Dominic Bagbio, Sheila Owusu, Stephen Tabiri (Tamale Teaching Hospital, Ghana), Dayang Anita Abdul Aziz (UKM Medical Centre, Kuala Lumpur, Malaysia).

**Contributors** The principal investigator (NW) conceived the idea for the study, gained study funding, wrote the study protocol, designed the data collection tools, established the study team, co-ordinated the pilot study, revised the study design/data collection tools following feedback and made critical revisions to the manuscript for publication. The steering committee contributed critical input and revisions to the funding application, study design, protocol and manuscript for publication. The writing committee drafted the protocol manuscript for publication and contributed as organising committee members. The organising committee assisted in the recruitment of and communication with collaborators to participate in the pilot study, helped to co-ordinate the pilot study and summarise the feedback, made critical revisions to the data collection tools in multiple languages and contributed to the study design. The lead investigators contributed to the study design and content of the data collection forms through feedback following participation in the pilot study. All contributed to the content of this manuscript.

**Funding** NW, Principal Investigator, is funded by the Wellcome Trust through a Clinical PhD in Global Health undertaken at King's College London (Funder Reference: 203905/Z/16/Z). The Wellcome Trust had no input into the study protocol other than to recommend open-access publication in a peer-reviewed journal and to make the anonymised dataset publicly available. Nick Sevdalis' (NS) research is supported by the National Institute for Health Research (NIHR) Collaboration for Leadership in Applied Health Research and Care South London at King's College Hospital NHS Foundation Trust. NS is a member of King's Improvement Science, which is part of the NIHR CLAHRC South London and comprises a specialist team of improvement scientists and senior researchers based at King's College London. Its work is funded by King's Health Partners (Guy's and St Thomas' NHS Foundation Trust, King's College Hospital NHS Foundation Trust, King's College London and South London and Maudsley NHS Foundation Trust), Guy's and St Thomas' Charity, the Maudsley Charity and the Health Foundation. NS and Andrew Leather are also supported by the NIHR Global Health Research Unit on Health System Strengthening in Sub-Saharan Africa, King's College London (GHRU 16/136/54) and by the ASPIRES research programme in LMICs (Antibiotic use across Surgical Pathways—Investigating, Redesigning and Evaluating Systems), funded by the Economic and Social Research Council. The views expressed are those of the authors and not necessarily those of the NHS, the NIHR or the Department of Health and Social Care.

**Competing interests** NS is the director of the London Safety and Training Solutions, which offers training in patient safety, implementation solutions and human factors to healthcare organisations. No other conflicts of interest are declared.

**Patient consent for publication** Not required.

**Ethics approval** This study has been classified as a clinical audit with written confirmation from King's College London Ethics Committee that it does not therefore require ethical approval. All participating centres have gained study approval to participate according to their local institutional ethical regulations.

**Provenance and peer review** Not commissioned; externally peer reviewed.

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
