## [Reviewer comments · BMJ Open]

ARTICLE DETAILS

TITLE (PROVISIONAL)	Management and Outcomes of Gastrointestinal Congenital Anomalies in Low-, Middle- and High-Income Countries: Protocol for a Multi-Centre, International, Prospective Cohort Study
AUTHORS	Wright, Naomi

VERSION 1 - REVIEW

REVIEWER	Saeed Dastgiri Director Tabriz Registry of Congenital Anomalies (TRoCA) School of Medicine Tabriz University of Medical Sciences Tabriz, Post Code: 5166615739, Iran
REVIEW RETURNED	27-Mar-2019

GENERAL COMMENTS	This is a research protocol for a multi-centre and international cohort study aiming to compare the management and outcomes of selected congenital anomalies between low-middle and high-income countries. Everything is almost clearly described as a protocol. It is acceptable for publication in BMJ Open.
--

REVIEWER	Mads Damkjær Hans Christian Andersen Children's Hospital, Denmark
REVIEW RETURNED	26-Apr-2019

GENERAL COMMENTS	Dear Mr/Mrs Overall I think that this is both an interesting and well planned out study. Publishing study protocols in full increases transparency, prevents duplication of studies and furthermore makes it easier to assess if the primary outcome has been changed after analyzing data (p-hacking). However, it is stated in the Instructions for reviewers that "If data collection is complete, we will not consider the manuscript". The authors state on Page 1 line 39 that the data collection will take place from October 2018 to April 2019. The data collection will therefore be complete before the manuscript is published. The manuscript can therefore not be considered for publication in BMJ Open. If, for some reason, this is a misunderstanding on my part of
---

	the guideline set to me by the journal, or if the data collection period is extended I shall be happy to revise my decision. Yours truly Mads Damkjær MD, PhD
--	---

REVIEWER	Sanjeev V. Thomas Sree Chitra Tirunal Institute for Medical Sciences, India
REVIEW RETURNED	21-May-2019

GENERAL COMMENTS	The authors have aimed to study the Management and Outcomes of Congenital Anomalies in Low-, Middle- and High-Income Countries: with a Protocol for a Multi-Centre, International, Prospective Cohort Study My main concern is that they are only examining 7 major malformations of the gastrointestinal system. There is no assessment of malformations of heart, nervous system, genito urinary system, or midline malformations such as cleft lip/palate etc. If the authors do not intend to study only the malformations pertaining to the GIT the same should reflect in the title and the text of the proposal. The term anomalies are generally reserved for minor derangements of development and malformations for those derangements that influence quality of life or require surgery. The authors may want to consider this change
---

REVIEWER	Dick Lindhout Utrecht University/University Medical Center Utrecht, the Netherlands
REVIEW RETURNED	02-Jun-2019

GENERAL COMMENTS	General remark: Excellent initiative. This study may provide preliminary results for further in-depth analyses. Specific remarks: 1. Page 3 (4/24, pdf 5/25), lines 8 and 16 In birth defects research, the terms incidence and prevalence are frequently used in ambiguous ways, whereas in most cases, prevalence is the correct term. Incidence requires the defining of the exposed cohort which is almost never possible for birth defects due to lack of a precise denominator (number of conceptuses), occurrence (genesis) of malformations at variable stages of pregnancy, and incomplete observation of (early) pregnancy losses. Therefore I suggest to use the term prevalence (at prenatal diagnosis, at birth, etc) instead of incidence, and, most importantly, thereby keeping the research group and future readers aware of these factors. Page 6 (7/24, pdf 8/25), lines 4-5 "Collaborators will enter anonymous data via the secure internet-based Research Electronic Data Capture (REDCap) system. This will be stored on King's College
---

	London REDCap server. No individual collaborator, institution or country will be independently identifiable in the presented or published results.." First of all, PATIENTS have to be listed as persons whose privacy needs protection! Not only collaborators, institutions or countries. Second, 'anonymous' doesn't guarantee that data cannot be traced back to individual patients (or collaborators, clinics or countries/regions). Current big data analysis and linkage with other data sources can easily lead to unwanted identification. There is a need for a legally binding protocol for access to, and usage and transfer of data etc. including site(s) of jurisdiction. Third, in case of multiple malformations, patients may enter the study sequentially in different institutions, even in different countries/regions. The in- and exclusion criteria for entry into the study do not address the possibility of multiple entry of the same patient but at different institutions or in different countries. International referrals for successive treatment of different malformations are not uncommon in case of complex cases: there may be a need to identify different CRFs belonging to the same patient. Page 6 (7/24, pdf 8/25), Line 46/47 "Vaginal delivery includes those requiring forceps and ventouse" Suggestion: Make also distinction between forceps and ventouse, since use of these are related to outcome, by indication and by effect they may have. There are 8 types of vaginal delivery: 2 (spontaneous or induced) x 4 (without, ventouse, forceps, ventouse and forceps) = 8. Hereby assuming that 'induced' means induction of labour. Final comment: Since this is a global study, I suggest to use and check definition and naming of countries and regions according to WHO.
--	---

VERSION 1 – AUTHOR RESPONSE

	Reviewer's Comment	Response	Section of the manuscript
Reviewer 1 – no revisions required			
Reviewer 2			
2	Question regarding eligibility of the study protocol for publication.	Following discussion with the BMJ Open editors, the study protocol is eligible for publication since the data collection is still ongoing until the end of July 2019.	Study design.
Reviewer 3			
3	My main concern is that they are only examining 7 major malformations of the gastrointestinal system. There is no assessment of malformations of heart, nervous system, genito urinary system, or midline malformations such as cleft lip/palate etc. It the authors do not intend to study only the	The title and abstract have been revised accordingly. The introduction details and justifies our decision to focus on a selection of common congenital anomalies involving the gastrointestinal tract.	Title, abstract, introduction.

	malformations pertaining to the GIT the same should reflect in the title and the text of the proposal.		
4	The term anomalies are generally reserved for minor derangements of development and malformations for those derangements that influence quality of life or require surgery. The authors may want to consider this change.	Thank you for this suggestion and opportunity to consider the multiple terms used to describe this group of conditions. In the UK and Europe we do include these conditions under the umbrella term 'congenital anomalies'. This is evidenced by BAPS-CASS – the British Association of Paediatric Surgeons Congenital Anomaly Surveillance System, which specifically focusses on such conditions. Also, EUROCAT, European Surveillance of Congenital Anomalies. I have added a statement to reflect the different terms used in the introduction.	Introduction
Reviewer 4			
5	Page 3 (4/24, pdf 5/25), lines 8 and 16 In birth defects research, the terms incidence and prevalence are frequently used in ambiguous ways, whereas in most cases, prevalence is the correct term. Incidence requires the defining of the exposed cohort which is almost never possible for birth defects due to lack of a precise denominator (number of conceptuses), occurrence (genesis) of malformations at variable stages of pregnancy, and incomplete observation of (early) pregnancy losses. Therefore I suggest to use the term prevalence (at prenatal diagnosis, at birth, etc) instead of incidence, and, most importantly, thereby keeping the research group and future readers aware of these factors.	Thank you for this suggestion. The use of 'incidence and prevalence' was purposeful. The incidence of congenital anomalies amongst foetuses is higher in LMICs due to a higher rate of micronutrient deficiencies, infections during pregnancy and use of teratogenic agents resulting from lack of awareness of the risks. The prevalence is also higher in LMICs because of a lack of antenatal diagnosis prohibiting terminations for congenital anomalies unlike in HICs. I have added details regarding this in the introduction.	Introduction
6	Page 6 (7/24, pdf 8/25), lines 4-5 "Collaborators will enter anonymous data via the secure internet-based Research Electronic Data Capture (REDCap) system. This will be stored on King's College London REDCap server. No individual collaborator, institution or country will be independently identifiable in the presented or published results.."	1) my apologies for this oversight on page 6 within the 'Data Collection' section. Of note, 'patients' were included in this statement within the Ethics and Dissemination section. To avoid duplications within the text I have deleted this statement from the 'Data Collection' section – it remains included within the 'Ethics and Dissemination' section.	Ethics and Dissemination.

	First of all, PATIENTS have to be listed as persons whose privacy needs protection! Not only collaborators, institutions or countries. Second, 'anonymous' doesn't guarantee that data cannot be traced back to individual patients (or collaborators, clinics or countries/regions). Current big data analysis and linkage with other data sources can easily lead to unwanted identification. There is a need for a legally binding protocol for access to, and usage and transfer of data etc. including site(s) of jurisdiction. Third, in case of multiple malformations, patients may enter the study sequentially in different institutions, even in different countries/regions. The in- and exclusion criteria for entry into the study do not address the possibility of multiple entry of the same patient but at different institutions or in different countries. International referrals for successive treatment of different malformations are not uncommon in case of complex cases: there may be a need to identify different CRFs belonging to the same patient.	2) I have added the term 'de-identified' to the relevant sentence within the 'Data Collection' section. We have followed all ethical protocols and policies at King's College London and the participating centres and Data Transfer Agreements have been legally signed between institutions where required. 3) Only centres who provide the primary intervention following diagnosis are allowed to include the patient in the study to avoid such duplications. Centres who receive a patient with one of the congenital anomalies and then transfer to another centre are not permitted to include the patient in the study. Subsequent, follow-up surgery for complex cases is not included in the study – only the primary intervention. I have clarified this in the text.	Data collection. Ethics and Dissemination Patient inclusion and exclusion
7	Page 6 (7/24, pdf 8/25), Line 46/47 "Vaginal delivery includes those requiring forceps and ventouse" Suggestion: Make also distinction between forceps and ventouse, since use of these are related to outcome, by indication and by effect they may have. There are 8 types of vaginal delivery: 2 (spontaneous or induced) x 4 (without, ventouse, forceps, ventouse and forceps) = 8. Hereby assuming that 'induced' means induction of labour.	Thank you for this suggestion. We appreciate the 8 different types of delivery may affect neonatal outcomes. We are not aware of literature suggesting a difference in outcomes of the congenital anomalies being studied in relation to these different modes of delivery. We have minimised the granularity of the data collection as much as possible to minimise the time burden on the local data collection teams so that the study is feasible in all settings, including LMICs with overburdened clinical teams. We have included variables known to have the greatest effects on outcome amongst the conditions being studied. I have added this within the 'Strengths and Limitations' section of the manuscript.	Strengths and Limitations

8	Since this is a global study, I suggest to use and check definition and naming of countries and regions according to WHO.	Thank you – we have done this. All co-authors have had the opportunity to read and input into the manuscript to ensure their institution and country names are correct.	Supplementary File 2. Collaborators
---	---	---	--

VERSION 2 – REVIEW

REVIEWER	Sanjeev V. Thomas Sree Chitra Tirunal Institute for Medical Sciences
REVIEW RETURNED	18-Jun-2019

GENERAL COMMENTS	The authors have addressed the concerns raised by the reviewers and modified the manuscript satisfactorily
--

REVIEWER	Dick Lindhout University Medical Center Utrecht, the Netherlands
REVIEW RETURNED	03-Jul-2019

GENERAL COMMENTS	Thank to the authors for explaining the changes. My point with respect to difference between incidence and prevalence is not well understood. For incidence to be determined, the denominator of the cohort should be known, and this is seldom known for birth defects for reasons outlined in my initial review. Etiological factors like micronutrients, phase of pregnancy etc don't make the difference between incidence and prevalence. It is the lack of denominator for total number of conceptions (almost always unknown) that makes prevalence the only correct term. With respect to the current project, the team has to decide between two options: a) define the terms incidence and prevalence clearly and distinctively as they are intended to be used, OR b) accept that both terms are interchangeably used
---

VERSION 2 – AUTHOR RESPONSE

Dr Dick Lindhout - thank you for your suggestion and explanation. I have changed the manuscript accordingly - removing the term incidence and instead using just the term prevalence.